# Identification of Critical Nodes in Power Grid Based on Improved PageRank Algorithm and Power Flow Transfer Entropy

**Jinhui Zeng, Yisong Wu, Jie Liu \*, Dong He and Zheng Lan**

College of Electrical and Information Engineering, Hunan University of Technology, Zhuzhou 412007, China; wuyisong@stu.hut.edu.cn (Y.W.)
* Correspondence: 14393@hut.edu.cn; Tel.: +86-186-7339-2323

**Abstract:** Identifying critical nodes in the power grid is a crucial aspect of power system security and stability analysis. However, the current methods for identification fall short in fully accounting for the power transfer characteristics between nodes and the consequences of node removal on the security and stability of power grid operation. To enhance the effective and accurate identification of critical nodes in the power grid, a method is proposed. This method is based on improved PageRank algorithm and node-weighted power flow transfer entropy, referred to as IPRA-PFTE. Firstly, based on the power flow and equivalent impedance between nodes, and the introduction of virtual nodes, an improved PageRank algorithm is obtained. Then the node-weighted power flow transfer entropy is derived by considering the uniformity of the transfer power flow distribution in the system following the removal of a node. Finally, the importance of nodes is obtained by combining the improved PageRank algorithm with the node-weighted power flow transfer entropy. The method's effectiveness and accuracy are validated through simulation using the IEEE 39-bus example and subsequent comparison with existing methods.

**Keywords:** critical node identification; importance of nodes; improved PageRank algorithm; weighted flow transfer entropy

## 1. Introduction

The safe and stable operation of the power system faces challenges due to the continuous development, increasing size, and growing complexity of the modern power system and its grid structure [1,2]. In recent years, major blackouts, both domestically and internationally, have primarily resulted from the removal of critical nodes or lines due to failure or attack. This, combined with large-scale transfers of system power flow, leads to overloading or ultra-low-voltage operation of other nodes or branch circuits, causing chain failures in the power system [2–4]. These nodes, vulnerable to greater disasters from failure or attack, are also referred to as critical nodes. Reference [5] analyzes the underlying causes of many global blackouts, many of which are caused by attacks on critical nodes in the power grid, which helps people understand the importance of maintaining power grid stability and identifying critical nodes. Therefore, accurately and effectively identifying these critical nodes in the power system is crucial for ensuring the safe and stable operation of the power grid. This effort holds significant theoretical and practical importance in preventing large-scale power outages in the grid.

Currently, numerous findings have emerged from research on methods for identifying critical nodes in power grids. These methods can be primarily categorized from a modeling perspective.

The first set of methods focuses on the dynamic characteristics of the power grid. Its core involves utilizing time-domain simulation analysis with trend calculation to illustrate the development of grid chain faults, with the ultimate goal of identifying critical nodes within the power grid. Reference [6] assessed the vulnerability of cascading failure propagation components based on power flow entropy, considering both impact and consequence

aspects. Reference [7] proposed a probabilistic approach to electrical diffusion based on Monte Carlo simulation and stochastic DC optimal currents for identifying critical nodes in renewable energy grids. Reference [8] established a large outage fault set based on weighted tidal entropy and coupled branch median, which can accurately describe the dynamic process of grid faults. Reference [9] took a data-driven perspective, combining random matrix theory and entropy theory to construct an identification index for critical nodes in the power grid. These methods effectively capture the physical characteristics of grid components, ensuring high simulation accuracy, but there is a trade-off between identification accuracy and efficiency, which makes it difficult to meet the demands of online applications

The second method, grounded in complex network theory and emphasizing grid topology, introduces synthesis methods, whether single-indicator or multi-indicator, for identifying critical nodes from various perspectives. References [10,11] improved the indexes of degree, cohesion, betweenness, and average shortest path in complex networks and allocated the weight proportion of each index from the subjective and objective perspectives. Reference [12] explored the relationship between network size and robustness in terms of local, global, and tidal characteristics of the grid and developed an AC tidal chain fault model. Reference [13] employed a range of metrics to comprehensively assess the importance of node structure and state. These methods primarily delve into the study of topological structure parameters. Notably, the PageRank algorithm assesses a node's importance based on both the quantity and quality of nodes pointing to it. In recent years, there have been developments in an improved PageRank algorithm for identifying critical nodes or lines in the power grid by incorporating the electrical characteristics of the grid. In Reference [14], the PageRank algorithm was utilized to identify critical nodes, considering both the grid's topology and load importance. References [15,16] discussed the impact of the selection of the damping factor on the performance of the algorithm. Reference [17] proposed a node interaction model based on the disruption factor and introduced it into the PageRank algorithm. Reference [18] introduced virtual nodes connecting generator and load nodes, identifying critical nodes while considering the transmission and transfer characteristics of different node types. The impact of the power communication network on the grid was considered in Reference [19], showcasing high computational efficiency. The key to applying this method to power systems lies in constructing a model based on the grid's operating state to better align with its actual physical characteristics.

In summary, this paper introduces an innovative method, denoted as IPRA-PFTE, for identifying critical nodes within a power grid. This method integrates the enhanced PageRank algorithm with current transfer entropy, with a comprehensive consideration of the actual power grid operation and the imperatives of security and stability. To begin, the method starts by constructing a transmission transfer matrix based on the transmission transfer characteristics between nodes. This matrix relies on line currents and equivalent impedances between nodes, and it introduces virtual nodes to enhance the PageRank algorithm. Subsequently, the paper addresses the impact of node removal on the grid's safe and stable operation. It proposes the concept of weighted tidal transfer entropy for grid nodes, which, when combined with the improved PageRank algorithm, forms identification indices for discerning the critical nodes of the grid. Finally, the effectiveness and correctness of the proposed identification method are validated using the IEEE 39 bus system as a case study.

## 2. Power Grid Modeling and PageRank Algorithm

### 2.1. Complex Network Model of Power Grid

As shown in Figure 1, nodes in a complex network represent the power system's generator bus, load bus, and substation bus, while the transmission lines and transformer branches are denoted as the edges. This network incorporates the directional aspect of line power flow, assigning the power flow value of each line as the weight for the corresponding edge [18].

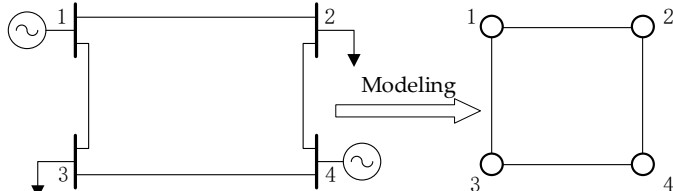

**Figure 1.** Modeling diagram.

Equation (1) illustrates the depiction of the power system in a simplified form as a directed weighted network:

$$D = (V, L, W) \tag{1}$$

where $V$ is the node set, $|V| = n$, $L$ is the edge set, $|L| = m$, and $W$ is the weight of the edge.

### 2.2. PageRank Algorithm

The PageRank algorithm stands as a pivotal algorithm within the framework of the Google search engine, originating from the collaborative efforts of Larry Page and Sergey Brin in 1996. At its essence, this algorithm hinges on the concept that a page's significance is contingent upon both the quantity and caliber of links directed towards it from other pages [19,20]. The iterative progression of the PageRank algorithm can be encapsulated as follows:

$$\boldsymbol{P}_{n+1} = \boldsymbol{G}\boldsymbol{P}_n \tag{2}$$

where $\boldsymbol{P}_n$ is the vector of PageRank (PR) values of each node obtained by the $n$th iteration; $\boldsymbol{G}$ is called the general transition matrix of a directed graph, which is an $n$-order square matrix, defined as:

$$\boldsymbol{G} = \alpha \boldsymbol{M} + \frac{(1 - \alpha)}{n} \boldsymbol{e}\boldsymbol{e}^T \tag{3}$$

where $\boldsymbol{M}$ is called the adjacency matrix; $\alpha$ is called the damping factor, Google initially set $\alpha$ to 0.85, which is a trade-off between the effectiveness of the algorithm and the rate of convergence of the power method [16]; $\boldsymbol{e}$ is an $n$-dimensional column vector with all 1.

If convergence is achieved in the iterative process, it is crucial that matrix $\boldsymbol{G}$ satisfy the conditions of being random, irreducible, and non-periodic. Once these three conditions are fulfilled, the application of the power law to the directed graph enables the derivation of a unique and positive steady-state PageRank vector.

## 3. Improved PageRank Algorithm

### 3.1. Problems and Improvements

As shown in Table 1, the power grid and the Internet share structural similarities, both being akin to weighted directed networks [17,21]. However, the power grid, being a more intricate system than the Internet, presents challenges when directly applying the (3) analysis to identify critical nodes. As shown in Figure 2, the standard PageRank algorithm is used to identify the critical nodes of the IEEE 39-bus system. The issues encountered include:

1.  The PageRank algorithm assumes users move randomly along the hyperlink matrix, distributing PageRank values to nodes they point to with equal probability. In contrast, the power system's network features varying interconnections between nodes, necessitating the allocation of PageRank values based on the degree of connection.
2.  When using the PageRank algorithm to assess node importance, the power nodes, serving as the network's starting points, have only outgoing lines and no incoming lines, while load nodes possess only incoming lines and no outgoing lines. This results in these two node types being assigned low importance, deviating from the actual dynamics of the power grid.

**Table 1.** Comparison of the web and power grid topology.

| Power Grid | Web | Weighted Directed Networks |
|---|---|---|
| Bus | Webpage | Node |
| Transmission line | Hyperlink | Edge |
| Power flow on the line | The probability of accessing hyperlinks | The weight of edge |
| Nodal load capacity | Visits to webpage | Load of node |
| Importance of a node | Initial quality of a webpage | Initial quality of a node |

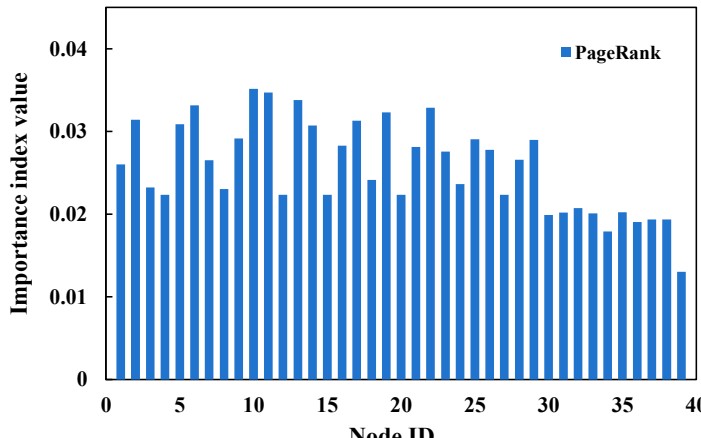

**Figure 2.** The identification results of standard PageRank.

To address these challenges, this paper enhances the PageRank algorithm in the following ways:

1. Modification of the general transfer matrix **G** based on power flow in the line and equivalent impedance between nodes. This adjustment creates an electrical transmission transfer matrix that aligns with the link characteristics of the power system.

2. Introduction of a virtual node $v$ within the original grid structure (Figure 3). To maintain power balance after incorporating virtual nodes, they are connected to all power generation nodes, and all load nodes are linked to the virtual nodes. Ultimately, the sizes of the load and generator output serve as the weights of the edges.

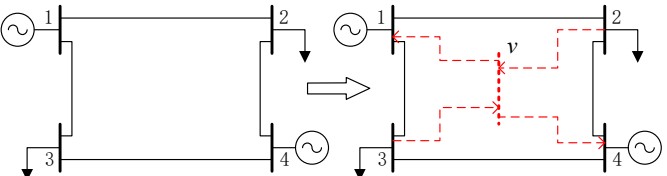

**Figure 3.** Introducing virtual nodes.

### 3.2. Electrical Adjacency Matrix

Upon the incorporation of power flow and the addition of a virtual node into the intricate network model of the power grid, the $n + 1$ of adjacency matrix **M** is listed as the size of all loads, and $n + 1$ is the output of all generator nodes. From this, the extended matrix $\overline{M}$ of the adjacency matrix **M** can be obtained. In this paper, it is referred to as the electrical adjacency matrix, with its elements defined as follows:

1. When $i = j$,

$$\overline{M}_{ij} = 0 \tag{4}$$

2. When $i \neq j$ there are three cases:

- When $0 < i < n, 0 < j < n$,

$$\overline{M}_{ij} = \begin{cases} \alpha_{ij} \dfrac{S(i,j)}{\sum\limits_{k \in O(i)} S(i,k)}, S_{load}(i) = 0 \\ \alpha_{ij} \dfrac{S(i,j)}{\sum\limits_{k \in O(i)} S(i,k) + S_{load}(i)}, S_{load}(i) \neq 0 \end{cases} \tag{5}$$

- When $0 < i < n + 1, j = n + 1$,

$$\overline{M}_{ij} = \frac{S_{load}(i)}{\sum\limits_{i=1}^{n} S_{load}(i)} \tag{6}$$

- When $i = n + 1, 0 < j < n + 1$,

$$\overline{M}_{ij} = \frac{S_{gen}(j)}{\sum\limits_{j=1}^{n} S_{gen}(j)} \tag{7}$$

In (4)–(7), the $\alpha_{ij}$ judgment function is 1 when node $i$ points to node $j$; otherwise, it is 0; $S(i, j)$ is the power flow value on the transmission line from node $i$ to node $j$; $S_{load}(i)$ is the load size on node $i$; $O(i)$ is the set of outgoing lines of node $i$; and $S_{gen}(j)$ is the generator output of node $j$.

### 3.3. Electrical Self-Link Matrix

To create the self-linking matrix $E$, the equivalent impedance between grid nodes is introduced, and the reciprocal of the equivalent impedance between each pair of nodes serves as the probability of information transmission between them. The elements in the self-linking matrix $E$ are defined as follows:

$$\begin{aligned} E_{ij} &= \frac{e(i,j)}{\sum\limits_{h=1}^{n} e(i,h)} \\ e(i,j) &= \frac{1}{d(i,j)} \end{aligned} \tag{8}$$

where $d(i, j)$ represents the equivalent impedance value between node $i$ and node $j$.

Equation (8) clearly indicates that a smaller equivalent impedance between node $i$ and node $j$ results in a larger $E_{ij}$ value, signifying a stronger and more robust coupling relationship between the two nodes. Therefore, the electrical link matrix based on the reciprocal of the equivalent impedance can characterize the possible probability of information transmission between any two nodes in the power grid. Introducing the $n$-dimensional column vector $E_v$ facilitates the derivation of the extended matrix $\overline{E}$ from the self-linking matrix $E$. This extended matrix is denoted as the electrical self-linking matrix in this paper. The $n \times 1$-dimensional column vector $E_v$ is introduced, and the extended matrix $\overline{E}$ of the self-linking matrix $E$ is obtained. In this paper, it is called the electrical self-linking matrix, as shown in Equation (9):

$$\overline{E} = \begin{bmatrix} E & E_v \\ E_v^T & 0 \end{bmatrix} \tag{9}$$

where

$$E_v(i) = \frac{1}{n} \tag{10}$$

### 3.4. Electrical Transmission Transfer Matrix

Based on the above electrical adjacency matrix $\overline{M}$ and electrical self-linking matrix $\overline{E}$, the electrical transmission transfer matrix $\overline{G}$ can be obtained.

$$\overline{G} = \left(\alpha\overline{M} + (1-\alpha)\overline{E}\right)^T \tag{11}$$

The electrical transmission transfer matrix presented in Equation (11) encompasses both the real information transmission transfer ratio between nodes in the power grid and adjacent nodes, as well as the potential information transmission transfer ratio between nodes and all other nodes in the network. Hence, the enhancement of the PageRank algorithm is approached from the standpoint of information transmission transfer:

$$\begin{bmatrix} R^{k+1} \\ R_v^{k+1} \end{bmatrix} = \overline{G} \begin{bmatrix} R^k \\ R_v^k \end{bmatrix} \tag{12}$$

Equation (12) can be written as:

$$\begin{bmatrix} R^{k+1} \\ R_v^{k+1} \end{bmatrix} = \overline{G}^k \begin{bmatrix} R^0 \\ R_v^0 \end{bmatrix} \tag{13}$$

In Equations (12) and (13), $R^k$ and $R_v^k$ denote the PageRank values associated with each node and virtual node in the power grid after the $k$th iteration. $R^0$ and $R_v^0$ represent the initial importance of each node and virtual node in the network, respectively.

As indicated in Equation (13), the choice of the initial value plays a role in influencing the iterative process. Therefore, this paper considers the electrical in-degree centrality and sets the initial value of each node:

$$\begin{aligned} R^0(i) &= \frac{1}{n-1} \sum_{j=1}^{n} \beta_{ij} S(j,i) \\ R_v^0 &= \frac{1}{n} \sum_{i=1}^{n} R^0(i) \end{aligned} \tag{14}$$

where $\beta_{ij}$ is the judgment function. When node $j$ points to node $i$, $\beta_{ij} = 1$, and $\beta_{ij} = 0$ in other cases. $S(j,i)$ is the power flow value on the transmission line from node $j$ to node $i$.

After the $k$th iteration, assuming convergence of the iterative process, the post-convergence PageRank vector entails eliminating the PR value associated with the virtual node. Following that, the remaining nodes' PR values undergo normalization, leading to the formation of vector $\overline{R}$. This vector indicates the relative importance of each node in the power grid, as demonstrated in Equation (15).

$$\overline{R}(i) = \frac{R^k(i)}{\sum_{i=1}^{n} R^k(i)} \tag{15}$$

The matrix $\overline{G}$ constructed shares similar properties with $G$ in the general definition of PageRank:

1.  Randomness: $\overline{G}$ is a random matrix.

**Proof.** Formed through the linear combination of random matrices $\overline{M}$ and $\overline{E}$, matrix $\overline{G}$ satisfies the condition $\forall j = 1, 2, 3, \ldots, n, n+1 \sum_{i=1}^{n+1} \overline{g}_{ij} = 1$ for its elements $\overline{G}_{ij}$. Therefore, it can be concluded that matrix $G$ is a random matrix. $\square$

2.  Irreducibility: $\overline{G}$ is an irreducible matrix.

**Proof.** Equations (3) through (9) reveal that the matrix $\overline{G}$ is non-negative, implying the strong connectivity of the directed graph associated with the matrix. Therefore, the matrix $\overline{G}$ is characterized as irreducible. □

3. Aperiodicity: $\overline{G}$ is a non-periodic matrix.

**Proof.** Because the matrix $\overline{G}$ is non-negative, and, when $k > 0$, there is $\overline{G}^k > 0$, so the matrix $\overline{G}$ is a non-periodic matrix. □

The aforementioned properties serve as evidence that the matrix $\overline{G}$ possesses a distinctive and positive steady-state PageRank vector.

## 4. Identification of Critical Nodes

### 4.1. Weighted Power Flow Transfer Entropy of Grid Node

Entropy serves as a gauge of the system's chaotic and disordered state. The focus of the weighted power flow transfer entropy (PFTE) in power grids is mainly on pinpointing critical lines in the grid. This measure characterizes the uneven distribution of transferred power flows among the remaining lines following the disconnection of a particular line [6]. Based on this concept, the PFTE for nodes in the power grid is defined in this paper as follows:

$$
\begin{aligned}
H(i) &= -\sum_{k=1}^{n} \mu_i(k)\varepsilon_i(k)\ln\varepsilon_i(k) \\
\mu_i(k) &= \frac{1}{|O(k)|}\sum_{j\in O(k)} F(k,j) \\
\varepsilon_i(k) &= \frac{\Delta P_{ki}}{\sum_{k=1}^{n}\Delta P_{ki}}
\end{aligned}
\tag{16}
$$

where $\mu_{ki}$ represents the average load rate on the outgoing line of node $k$ when node $i$ is excluded; $O(k)$ denotes the set of outgoing lines from node $k$; $F(k,j)$ signifies the load rate on the line from node $k$ to node $j$; $\varepsilon_{ki}$ represents the impact ratio of power flow transfer to node $k$ following the removal of node $I$; and $\Delta P_{ki}$ represents the increase in transmission capacity allocated to node $k$ after the removal of node $i$, that is:

$$
\Delta P_{ki} = P_{ki} - P_{k0}
\tag{17}
$$

where $P_{k0}$ is the initial transmission capacity of node $k$, and $P_{ki}$ is the transmission capacity of node $k$ after node $i$ is removed.

In this paper, the normalized index $\overline{H}$ is called the PFTE for power grid nodes, that is:

$$
\overline{H}(i) = \frac{H(i)}{\sum_{i=1}^{n} H(i)}
\tag{18}
$$

where $\overline{H}(i)$ signifies the evenness of the distribution of transfer power flow following the removal of node $i$. A higher $\overline{H}(i)$ implies a more uneven distribution of transfer power flow after the removal of node $i$, which, in turn, contributes to heightened power flow fluctuations within the power grid. As a result, the PFTE for power grid nodes functions as an evaluative metric for assessing the significance of nodes for ensuring the secure and stable operation of the power grid.

### 4.2. Calculation Method of IPRA-PFTE

In assessing node importance, the enhanced PageRank algorithm takes into account the topological information transmission characteristics and the power flow state between nodes. On the other hand, the PFTE for power grid nodes considers the fluctuation in power flow and the uniformity of transfer power flow arising from the remaining nodes

after the removal of a particular node. By amalgamating these two facets, a novel method, referred to as IPRA-PFTE, is introduced in this paper for discerning critical nodes within the power grid. This approach yields a fresh vector for assessing node importance, and the corresponding calculation method is elucidated in Equation (19):

$$\overline{B}(i) = \omega_1 \overline{R}(i) + \omega_2 \overline{H}(i) \tag{19}$$

where $\omega_1$ is the weight share of the PageRank vector and $\omega_2$ is the weight share of the PFTE for power grid nodes, which satisfies $\omega_1 + \omega_2 = 1$.

Figure 4 illustrates the process of IPRA-PFTE for identifying critical nodes within the power system.

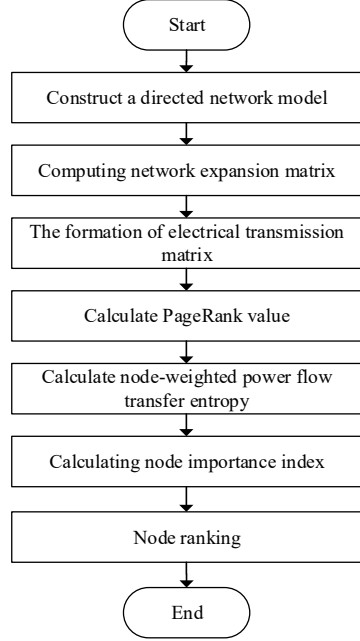

**Figure 4.** Flow chart of calculation.

Step 1: The weighted directed network model of the power grid is constructed using Equation (1).
Step 2: The network expansion matrix is calculated using Equations (4)–(10).
Step 3: The network electrical transmission transfer matrix is calculated using Equation (11).
Step 4: The vector $\overline{R}$ is calculated using Equations (13)–(15).
Step 5: The node-weighted power flow transfer entropy $\overline{H}$ is calculated using Equations (16)–(18).
Step 6: The vector $\overline{B}$, which is an indicator for evaluating the importance of the node, is calculated using Equation (19).
Step 7: The nodes are sorted according to the size of the evaluation index.

## 5. Simulation and Discussion

### 5.1. Critical Node Identification Results

In this paper, the critical nodes within the system are identified, and the effectiveness of IPRA-PFTE is validated using the IEEE 39-bus system as a test case. The system's network topology is illustrated in Figure 5, comprising 10 generator nodes and 46 transmission lines. Detailed data is extracted from the MATPOWER 7.0 dataset and simulated using MATLAB 2018b.

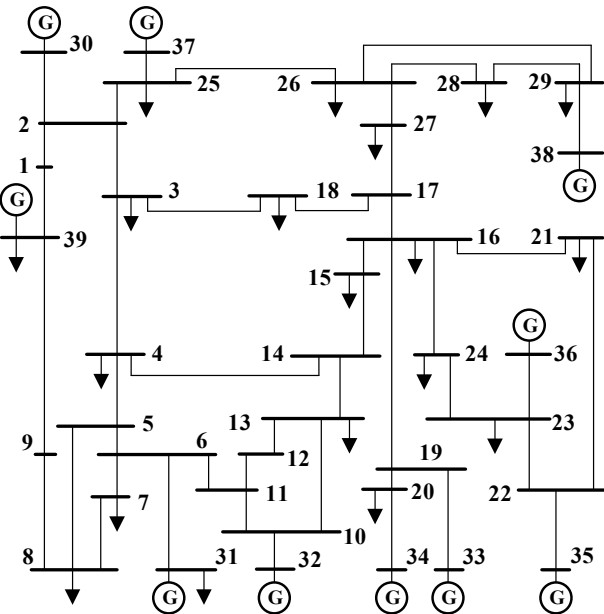

**Figure 5.** The schematic diagram of IEEE 39-bus system.

Figure 6 illustrates a comparison between the results of the improved PageRank algorithm and the PFTE for power grid nodes. It can be seen that there is a large difference between the two identification results. This is because the improved PageRank algorithm identifies critical nodes from the perspective of transmission characteristics between nodes, and the weighted power flow transfer entropy identifies critical nodes from the perspective of power flow transfer distribution generated after node removal. Therefore, this paper integrates the features of both to derive IPRA-PFTE, and Figure 7 presents its identification results.

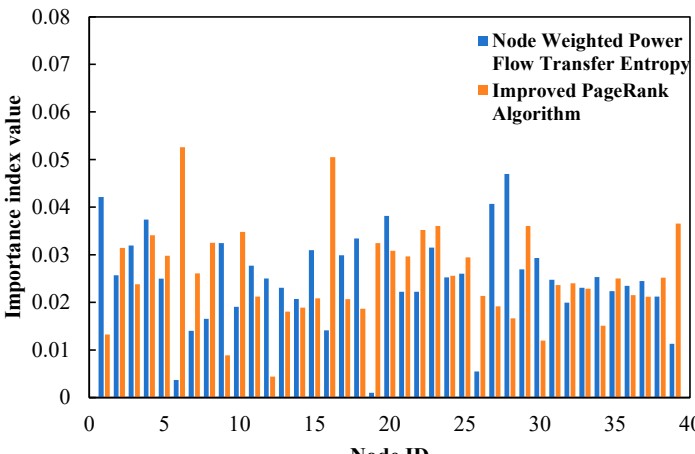

**Figure 6.** Improved PageRank algorithm and node-weighted power flow transfer entropy identification results.

In this study, the IPRA-PFTE algorithm is contrasted with three significant node algorithms: a fusion of random matrix theory and entropy theory [9], the ETPD-CNIA algorithm that integrates topology and power flow distribution [17], and the improved PageRank algorithm, which accounts for the transmission characteristics of various node types [18]. The ranking results of the four algorithms on the IEEE 39-bus system are presented in Table 2.

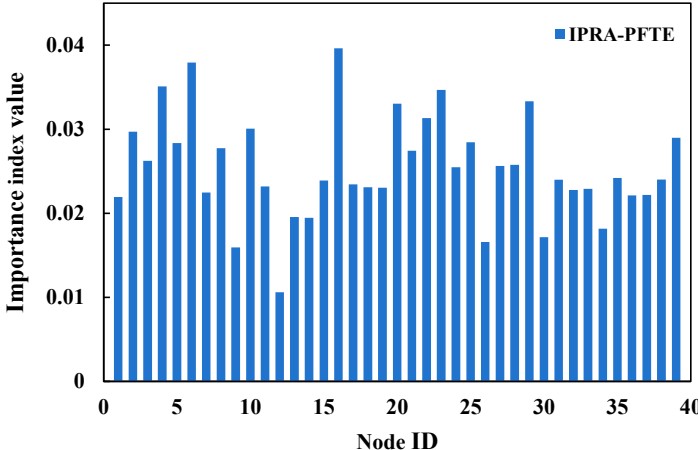

**Figure 7.** The identification results of IPRA-PFTE.

**Table 2.** Comparison of critical node identification results.

| Rank | IPRA-PFTE | Random Matrix &Entropy [9] | ETPD—CNIA [17] | Improved PageRank [18] |
|---|---|---|---|---|
| 1 | 16 | 6 | 16 | 16 |
| 2 | 6 | 5 | 6 | 8 |
| 3 | 4 | 11 | 29 | 4 |
| 4 | 23 | 8 | 26 | 6 |
| 5 | 1 | 10 | 31 | 26 |
| 6 | 29 | 7 | 19 | 9 |
| 7 | 20 | 13 | 22 | 5 |
| 8 | 22 | 14 | 21 | 14 |
| 9 | 10 | 4 | 20 | 29 |
| 10 | 2 | 15 | 10 | 3 |

The critical node findings from the IPRA-PFTE identification align closely with those of other methods, with the critical nodes set largely overlapping with results from alternative approaches. For instance, Node 16 links five nodes (15, 17, 19, 21, and 24), and the cumulative absolute power values on the connected lines reach 1680 MW. This places it second only to Node 6, which, with 1926 MW, serves as a pivotal hub for generator power transmission. Notably, the removal of Node 16 would lead to the grid's segmentation into three islands, significantly impacting both the safety and stability of power grid operations and its overall topology. This observation is consistent with the findings in References [16,17]. While Node 6 may not hold the same topological significance as Node 16, it stands out due to having the highest sum of absolute power values on connected lines among all nodes, marking it as a crucial node. Nodes 23, 29, 20, 22, 10, and 2 represent power transmission nodes for generators 36, 38, 34, 35, 32, and 30, respectively. The removal of these nodes would disrupt generator power transmission, leading to system power imbalance. Additionally, Nodes 4 and 1 serve as essential load nodes, and their removal would result in highly uneven power flow redistribution to other nodes within the power grid.

In summary, this paper demonstrates the accuracy of the proposed IPRA-PFTE algorithm by comparing it with existing methods and analyzing the position of nodes in the network structure and their inherent properties.

*5.2. Network Performance Impact Analysis*

To further illustrate the effectiveness and superiority of the IPRA-PFTE algorithm, the first 10 critical nodes obtained by IPRA-PFTE and the other three algorithms are attacked in descending order in this paper. In observing the changes in network performance

indicators during the attack process and the network performance indicator values after the completion of the attack on all 10 nodes, the faster the downward trend of the indexes and the smaller the final value, the better the critical node identification effect and the better the algorithm performance.

5.2.1. Network Performance Indicators

This paper employs network transmission efficiency [22], load survival rate [23], and comprehensive performance indicators to measure the performance of each algorithm from different perspectives. The specific definitions of the three are as follows:

1. Network Transmission Efficiency

Changes in the power grid resulting from the removal of nodes or transmission lines impact the power transmission path and capacity of the grid. Consequently, network transmission efficiency is employed to characterize the overall transmission capacity alteration following a grid attack, providing a measure of the importance of the attacked nodes. The network transmission efficiency, denoted as $\eta$, is defined as follows:

$$\eta = \frac{E_{PS\_k}}{E_{PS\_0}} \times 100\% \tag{20}$$

where $E_{PS\_0}$ is the network effectiveness in the initial state of the power grid, and $E_{PS\_k}$ is the network effectiveness after the power grid has suffered $k$ faults. The network effectiveness is calculated, as shown in Equation (21):

$$E_{PS} = \frac{1}{n(n-1)} \sum_{i \in S, j \in L} \frac{1}{D_{ij}} \tag{21}$$

where $S$ and $L$ are the sets of generator and load nodes, respectively, and $D_{ij}$ is the shortest electrical distance between nodes in the power grid. The reduction in the value of $\eta$ signifies a greater impact on the grid's power transfer capability when a node is attacked. This indicates that the transmission path between each pair of generators and the load node is lengthened after an attack, highlighting the increased importance of the node.

2. Load Survival Rate

Ensuring the regular power demand of users is a fundamental function of the power system. Consequently, the significance of attacked nodes can be assessed through load loss. The load survival rate $LS$ is calculated as shown in Equation (22):

$$LS = \frac{L_k}{L_0} \times 100\% \tag{22}$$

The initial load of the power grid is denoted as $L_0$, and the remaining load after the $k$th attack is represented by $L_k$. The significance of the attacked node increases as the $LS$ decreases.

3. Comprehensive Performance Indicator

This paper introduces two indicators, load survival rate and network transmission efficiency. However, nodes may exert a greater impact on one indicator following an attack. Consequently, utilizing the load survival rate and network transmission efficiency, the paper introduces a comprehensive performance index (CPI). This index is defined as the weighted sum of the two indicators:

$$CPI = \lambda_1 LS + \lambda_2 \eta \tag{23}$$

where $\lambda_1$ is the weight share of load survival rate and $\lambda_2$ is the weight share of network transmission efficiency and satisfies $\lambda_1 + \lambda_2 = 1$. In this context, a value of 0.5 is assigned.

The greater the rate of decrease in the comprehensive performance index, the more crucial the attacked node becomes.

5.2.2. Comparison Analysis

Figure 8 shows that the ETPD-CNIA and Improved PageRank algorithms perform as well as IPRA-PFTE and significantly better than the Random Matrix&Entropy in the network transmission efficiency index after the initial three nodes have been attacked.

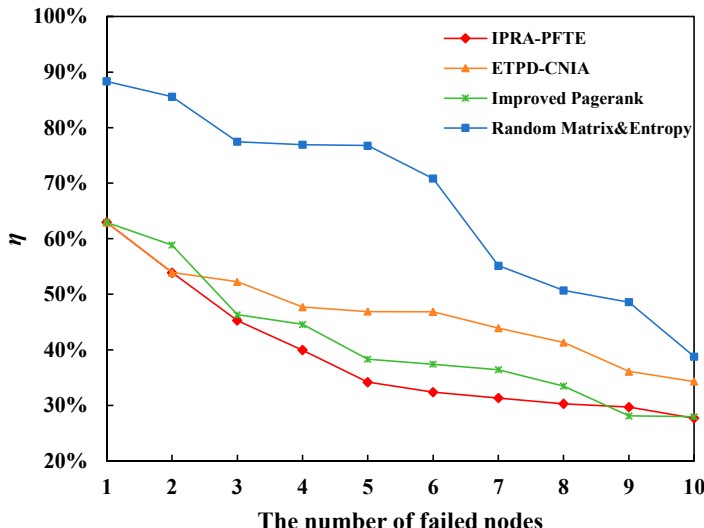

**Figure 8.** Comparison of network transmission efficiency.

Random Matrix&Entropy are grounded in statistical characteristics and the physical attributes of the power grid. Consequently, the decreasing trend of n is not very obvious after the current attack on the five critical nodes. Both ETPD-CNIA and the improved PageRank algorithm identify critical nodes by assessing power flow distribution. After removing the initial three nodes, there is an increased likelihood of cascading failures and power grid splitting being triggered, bringing their performance closer to IPRA-PFTE in the early part of the curve in Figure 8. Nevertheless, the distinct advantage of IPRA-PFTE becomes increasingly evident as the number of attacked critical nodes escalates. Upon the removal of all critical nodes, the final network transmission efficiency is 27.71%.

Figure 9 illustrates that there is a significant decrease in the load survival rate when the critical nodes are attacked by the IPRA-PFTE algorithm in the case of the load survival rate metric. The curve of IPRA-PFTE decreases faster, and the final load survival rate reaches 25.61% when all the critical nodes are removed. After the first five critical nodes identified by the improved PageRank algorithm and Random Matrix&Entropy are removed, the load survival rate decreases slowly. This is because the two identify critical nodes from the perspective of node voltage data mining and the power flow distribution characteristics between nodes. The final load survival rates after all critical nodes are removed are 39.98% and 52.81%, respectively. ETPD-CNIA identifies critical nodes based on the output power of the generator, so its performance under this index is not as good as IPRA-PFTE. The final load survival rate, after all critical nodes are removed, is 34.35%. IPRA-PFTE significantly outperforms the other three algorithms under this metric, both in terms of downward trend and final value.

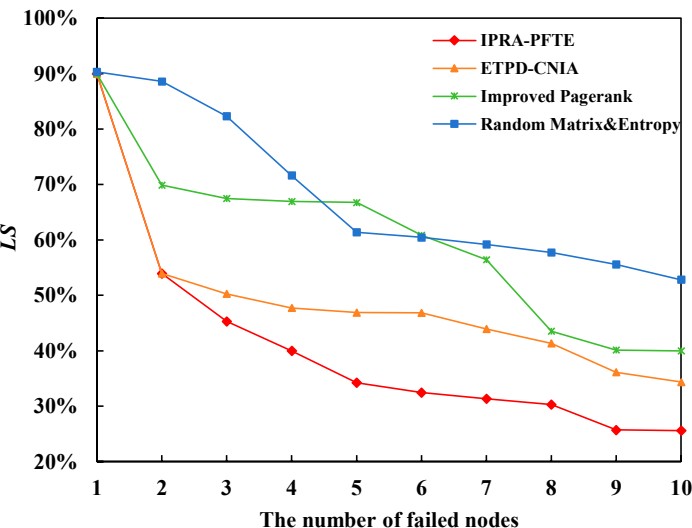

**Figure 9.** Comparison of load survival rate.

IPRA-PFTE considers not only the transmission transfer characteristics of the system nodes but also takes into account the power flow distribution of the system after node removal. This is equivalent to simultaneously considering the load survival rate and network transmission efficiency, optimizing its comprehensive performance. Upon the removal of all critical nodes, the comprehensive performance index is 26.65%, as depicted in Figure 10.

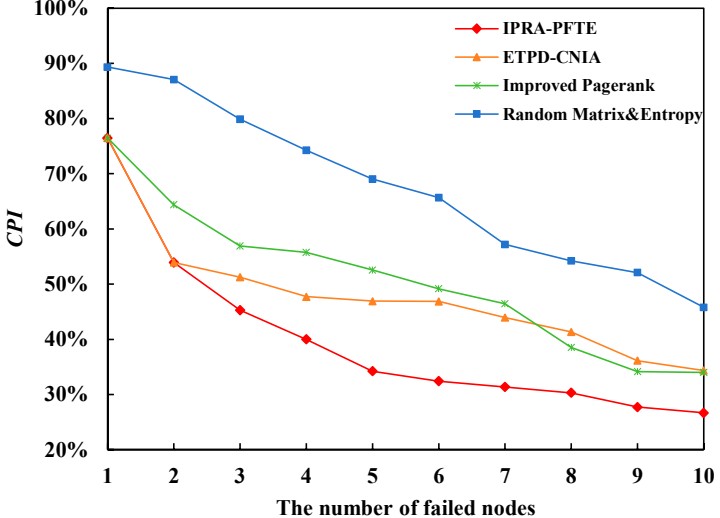

**Figure 10.** Comparison of comprehensive performance indicators.

To sum up, the IPRA-PFTE algorithm considers the transmission and transfer characteristics of nodes, the network topology, and the uniformity of system power flow distribution after node removal. Based on the alterations in the aforementioned three performance indicators and corresponding numerical analyses after the removal of all critical nodes, it can be seen that IPRA-PFTE can effectively and accurately identify the critical nodes of the power grid.

## 6. Conclusions

Considering the complex operations of power systems, the identification of critical nodes in the power grid is essential for ensuring its safe and stable operation. This paper introduces a critical node identification algorithm, IPRA-PFTE, which takes into account

the link relationships, transmission transfer characteristics between nodes, and uniformity of system power flow distribution after node removal.

Firstly, using the power flow information of the line and the equivalent impedance between the nodes, this paper creates an electrical transmission matrix. This matrix elucidates the degree of interaction among the nodes. This process resulted in an improved PageRank algorithm. Subsequently, considering the power flow fluctuations caused by other nodes in the system after node removal and the uniformity of power flow distribution, this paper proposes a weighted power flow transmission entropy of the nodes. Ultimately, the critical nodes are identified by weighting and summing the crucial new indicators obtained from the initial two calculations.

Through simulating the IEEE 39-bus system and conducting a comparative analysis with three other algorithms, the performance of IPRA-PFTE is evaluated under three network performance indicators. The results show that the IPRA-PFTE algorithm is significantly better than the other algorithms. In the actual project, monitoring or taking measures to strengthen the identified critical nodes of the power grid will provide practical guidance for the safe and stable operation of the power system as well as the prevention and reduction of large-scale power outages.

The method proposed in this paper has not yet taken into account the effect of the stochastic output characteristics of new energy sources on the identification of critical nodes. Consequently, the upcoming research will prioritize the identification of critical nodes in high-permeability new energy systems.

**Author Contributions:** Conceptualization, J.Z. and Y.W.; methodology, J.Z. and Y.W.; software, Y.W. and J.L.; validation, J.L.; formal analysis, Y.W.; investigation, Y.W. and J.L.; resources, J.Z.; data curation, D.H.; writing—original draft preparation, Y.W.; writing—review and editing, Y.W. and J.L.; visualization, J.Z. and Y.W.; supervision, J.Z. and D.H.; project administration, Z.L.; funding acquisition, J.Z. All authors have read and agreed to the published version of the manuscript.

**Funding:** This research was funded by the National Natural Science Foundation of China, grant number 52377185, and the Scientific Research and Innovation Foundation of Hunan University of Technology, grant number CX2311.

**Data Availability Statement:** The data used to support the research results of this paper are included within the article.

**Conflicts of Interest:** The authors declare no conflicts of interest.

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
