# Peer review of "Identification of Critical Nodes in Power Grid Based on Improved PageRank Algorithm and Power Flow Transfer Entropy"

_electronics, doi:10.3390/electronics13010184_

Round 1
Reviewer 1 Report
Comments and Suggestions for Authors
My biggest concern is related to the literature overview. The authors reused many expressions from the existing literature and did not cite them. This should be corrected.
After eq. 3 please briefly explain why the parameter alpha typically takes the value of 0.85.
Check if the position of figures should be left or centralized.
Comments on the Quality of English LanguageThe paper is well-written, with minor typing errors and grammar mistakes.
- I noticed that sometimes there is a small instead of a capital letter, for example:
3.1 problems and improvements
node ranking |
in Fig2
- This is not the correct sentence:
Where omega1 omega2 are the weight ratios of PageRank vector and node weighted power flow transfer entropy, respectively, satisfying ...
- Also, this does not sound right:
Finally The importance of nodes is calculated by combining... (in abstract)
Read the paper again and correct these small errors (and maybe others that I have missed).
Author Response
Dear reviewers and editor,
Firstly, the authors would like to thank Editor, Co-EIC, Associated Editor, and all reviewers for critically reviewing the paper and pointing out the ways to improve the technicality and readability aspects of this paper. Besides, we sincerely appreciate your consideration of our manuscript. We have revised the manuscript according to the review reports. All the changes in the manuscript have been highlighted in red color.
- My biggest concern is related to the literature overview. The authors reused many expressions from the existing literature and did not cite them. This should be corrected.
Reply: Thanks for reviewer’s advice. This has been corrected in the manuscript, where references have been marked in Chapter 2, which describes the underlying theory of grid modeling and PageRank algorithms, and in Chapter 4, which describes the underlying theory of entropy for tidal transfer, among others.
- After eq. 3 please briefly explain why the parameter alpha typically takes the value of 0.85.
Reply: Thanks for reviewer’s advice. In Google's PageRank model, the parameter alpha is used to control the proportion of the Internet's own hyperlink structure that the model assigns to the Internet, and as alpha approaches 1, the greater the proportion of the hyperlink structure.Google initially set alpha to 0.85, which can be viewed as a trade-off between the effectiveness of the algorithm and the rate of convergence of the power law. On the one hand, if the value of alpha is small, it means that more consideration is given to the random behavior of surfers, which is obviously unreasonable; on the other hand, if the value of alpha is large, the convergence rate of the power law will be reduced, and when alpha is too close to 1, it will also lead to some important web page sorting confusion[16]. This is briefly explained in the manuscript.
- Check if the position of figures should be left or centralized.
Reply: Thanks for reviewer’s advice. We carefully checked the position of the graph. By comparing the template downloaded from the official website of the journal with the published articles, the position of the graph is aligned on the left side and indented by 4.6 cm.
- I noticed that sometimes there is a small instead of a capital letter, for example: 3.1 problems and improvements “node ranking” in Fig2
Reply: Thank you for pointing out the errors. We have corrected the incorrect use of upper and lower case letters that you pointed out in the manuscript, as well as double-checked other locations for similar problems and corrected them.
- This is not the correct sentence: “ Where omega1 omega2 are the weight ratios of PageRank vector and node weighted power flow transfer entropy, respectively, satisfying ...”
Reply: Thank you for pointing out the errors. The sentence has been replaced with: “where omega1 is the weight share of the PageRank vector and omega2 is the weight share of the PFTE for power grid nodes, and satisfies omega1+ omega2=1” Other similar sentences in the manuscript have also been replaced accordingly.
- Also, this does not sound right: “Finally The importance of nodes is calculated by combining... (in abstract)”
Reply: Thank you for pointing out the errors. The sentence has been replaced with: “Finally, the importance of nodes is obtained by combining the improved PageRank algorithm with the node-weighted power flow transfer entropy.”
References
[6] Li, Y; Liu, J; Liu, X; et al. Vulnerability Assessment in Power Grid Cascading Failures Based on Entropy of Power Flow. Au-tomation of Electric Power Systems, 2012, 36(19): 11-16.
[15] Ma, Z; Liu, F; Shen, C; et al. Rapid Identification of Vulnerable Lines in Power Grid Using Modified PageRank Algo-rithm——Part I: Theoretical Foundation. Proceedings of the CSEE, 2016, 36(23): 6363-6370+6601.
[16] Ma, Z; Liu, F; Shen, C; et al. Rapid Identification of Vulnerable Lines in Power Grid Using Modified PageRank Algo-rithm——Part II: Factors Affecting Identification Results. Proceedings of the CSEE, 2017, 37(01): 36-45.
[17] Fan, B; Shu, N; Li, Z; Li, F. Critical Nodes Identification for Power Grid Based on Electrical Topology and Power Flow Dis-tribution. IEEE Systems Journal, vol. 17, no. 3, pp. 4874-4884, Sept. 2023,
[18] Zhu, D; Wang, R; Cheng, W; et al. Critical transmission node identification method based on improved PageRank algorithm. Power System Protection and Control, 2022,50(05): 86-93.
[19] Li, C; Kang, Z; Yu, H; et al. Identification Method of Key Nodes in Power System Based on Improved PageRank Algorithm. Transactions of China Electrotechnical Society, 2019, 34(09): 1952-1959.
Thanks again for your valuable advice!

Reviewer 2 Report
Comments and Suggestions for Authors
The paper presents a comprehensive and detailed approach to modeling power grids using complex network models and an improved PageRank algorithm. Here are some comments and suggestions for further refinement:
1. The paper does a commendable job of explaining the complex network model of the power grid. However, it could benefit from more detailed illustrations or examples to clearly demonstrate how nodes and edges are represented in this model. This could help in better visualizing the network structure, especially for readers less familiar with complex network theory.
2. The use of the PageRank algorithm is an innovative approach, but the paper could delve deeper into the rationale behind choosing this specific algorithm for power grid analysis. Discussing its advantages and potential limitations in this context would provide a more balanced view.
3. While the paper identifies challenges in applying the PageRank algorithm to power grids and proposes improvements, it might be beneficial to include a comparative analysis. This could involve comparing the results of the standard PageRank algorithm with the modified version in the context of power grid networks. Such a comparison could quantitatively demonstrate the effectiveness of the proposed modifications.
4. The paper could expand on the practical implications of this research. How can these models be applied in real-world scenarios? Discussing potential applications in power grid management, predictive maintenance, or optimization could enhance the paper's relevance to industry professionals.
5. Including case studies or simulations to validate the proposed models and algorithmic modifications would strengthen the paper. This could involve applying the model to a real or simulated power grid and analyzing its performance in identifying critical nodes or predicting network failures.
6. It would be enriching to see a comparison with other network models used in power grid analysis. How does this approach compare with other established methods in terms of accuracy, computational efficiency, and scalability?
7. The paper could benefit from a section discussing potential avenues for future research. This might include further algorithmic refinements, exploring different types of network models, or integrating this approach with other data-driven techniques like machine learning.
8. Ensure that the technical details, especially mathematical formulations, are both accurate and accessible. While the paper should maintain its rigorous scientific approach, simplifying complex concepts without losing their essence could make it more accessible to a broader audience.
Author Response
Dear reviewers and editor,
Firstly, the authors would like to thank Editor, Co-EIC, Associated Editor, and all reviewers for critically reviewing the paper and pointing out the ways to improve the technicality and readability aspects of this paper. Besides, we sincerely appreciate your consideration of our manuscript. We have revised the manuscript according to the review reports. All the changes in the manuscript have been highlighted in red color.
- The paper does a commendable job of explaining the complex network model of the power grid. However, it could benefit from more detailed illustrations or examples to clearly demonstrate how nodes and edges are represented in this model. This could help in better visualizing the network structure, especially for readers less familiar with complex network theory.
Reply: Thanks for reviewer’s advice. The following figure is an illustration we added in Section 2.1 of the manuscript. The IEEE 4-bus system is used as an example of modeling to illustrate the modeling process more intuitively.
Figure 1. Modeling diagram
- The use of the PageRank algorithm is an innovative approach, but the paper could delve deeper into the rationale behind choosing this specific algorithm for power grid analysis. Discussing its advantages and potential limitations in this context would provide a more balanced view.
Reply: Thanks for reviewer’s advice. We add a comparison table in Section 3.1 of the manuscript to compare the similarities and differences between the power grid, the Internet, and the weighted directed graph to analyze the basic principles of selecting the PageRank algorithm.
Table 1. Comparison of the web and power grid topology
Power grid |
Web |
Weighted directed networks |
Bus |
Webpage |
Node |
Transmission line |
Hyperlink |
Edge |
Power flow on the line |
The probability of accessing hyperlinks |
The weight of edge |
Nodal load capacity |
Visits ofwebpage |
Load of node |
Importance of a node |
Initial quality of a webpage |
Initial quality of a node |
- While the paper identifies challenges in applying the PageRank algorithm to power grids and proposes improvements, it might be beneficial to include a comparative analysis. This could involve comparing the results of the standard PageRank algorithm with the modified version in the context of power grid networks. Such a comparison could quantitatively demonstrate the effectiveness of the proposed modifications.
Reply: Thanks for reviewer’s advice. The following figure is an illustration we added in Section 3.1 of the manuscript. It describes the results obtained by using the standard PageRank algorithm to identify the critical nodes of the IEEE 39-bus system, which is used to illustrate the limitations of the critical node identification results obtained by using the standard PageRank algorithm.
Figure 2. The identification results of standard PageRank
- The paper could expand on the practical implications of this research. How can these models be applied in real-world scenarios? Discussing potential applications in power grid management, predictive maintenance, or optimization could enhance the paper's relevance to industry professionals.
Reply: Thanks for reviewer’s advice. The practical implications of this research is that in practical engineering, monitoring or taking measures to strengthen the identified critical nodes of the power grid will have practical guiding significance for the safe and stable operation of the power system and the prevention and reduction of large-scale blackouts. We also extend the practical significance of this study in the conclusion part of the manuscript , and the expansion part is highlighted in red..
- Including case studies or simulations to validate the proposed models and algorithmic modifications would strengthen the paper. This could involve applying the model to a real or simulated power grid and analyzing its performance in identifying critical nodes or predicting network failures.
Reply: Thanks for reviewer’s advice. Limited by the length of the manuscript, we will further improve the algorithm in the subsequent research and apply it to the actual regional power grid model.
- It would be enriching to see a comparison with other network models used in power grid analysis. How does this approach compare with other established methods in terms of accuracy, computational efficiency, and scalability?
Reply: Thanks for reviewer’s advice. The method in this paper is compared with other existing methods in the manuscript, and the results show that the accuracy of this method is better than other existing methods.
At the same time, since the structure of the IEEE 39-bus system is not particularly complex, each algorithm is able to obtain the results in a very short period of time, and therefore, the impact on the computational efficiency is very small in this case.
However, we consider further improving the algorithms in subsequent studies and comparing the computational efficiency and scalability with other algorithms when applied to more complex simulated systems as well as real power grids.
- The paper could benefit from a section discussing potential avenues for future research. This might include further algorithmic refinements, exploring different types of network models, or integrating this approach with other data-driven techniques like machine learning.
Reply: Thank you for your feedback on my article. In the last section of the conclusion, we discuss the potential research approaches for future research: “The method proposed in this paper has not yet taken into account the effect of the stochastic output characteristics of new energy sources on the identification of critical nodes. Consequently, the upcoming research will prioritize the identification of criti-cal nodes in high-permeability new energy systems.”, and your suggestions also provide us with a good research direction.
- Ensure that the technical details, especially mathematical formulations, are both accurate and accessible. While the paper should maintain its rigorous scientific approach, simplifying complex concepts without losing their essence could make it more accessible to a broader audience.
Reply: Thank you for your feedback on my article. We carefully examined the technical details in the manuscript to make sure there were no problems, and the original manuscript has simplified the complex concepts as much as possible.
References
[17] Fan, B; Shu, N; Li, Z; Li, F. Critical Nodes Identification for Power Grid Based on Electrical Topology and Power Flow Dis-tribution. IEEE Systems Journal, vol. 17, no. 3, pp. 4874-4884, Sept. 2023,
[19] Li, C; Kang, Z; Yu, H; et al. Identification Method of Key Nodes in Power System Based on Improved PageRank Algorithm. Transactions of China Electrotechnical Society, 2019, 34(09): 1952-1959.
Thanks again for your valuable advice!

Reviewer 3 Report
Comments and Suggestions for Authors
This paper proposed an algorithm for identifying critical nodes in the power grid with the consideration of the power transfer characteristics between nodes and the consequences of node removal on the security and stability of power grid operation.
1. Background part need to be modified to demonstrate the importance of the critical nodes identification. E.g., Paper “Adaptive Bidirectional Droop Control for Electric Vehicles Parking Lot with Vehicle-to-Grid Service in Microgrid” can help people know the importance of the grid stability. The proposed algorithm can also help paper “A hierarchical microgrid protection scheme using hybrid breakers” to locate the location of the breaker.
2. In the comparison table, please also compare the latency with other algorithms.
3. Please add more explanation for Fig.7. Especially the performance between the blue and green line.
4. Are there any drawbacks for introducing virtual nodes?
Comments on the Quality of English Languagenot bad
Author Response
Dear reviewers and editor,
Firstly, the authors would like to thank Editor, Co-EIC, Associated Editor, and all reviewers for critically reviewing the paper and pointing out the ways to improve the technicality and readability aspects of this paper. Besides, we sincerely appreciate your consideration of our manuscript. We have revised the manuscript according to the review reports. All the changes in the manuscript have been highlighted in red color.
Response to Reviewer 3
- Background part need to be modified to demonstrate the importance of the critical nodes identification. E.g., Paper “Adaptive Bidirectional Droop Control for Electric Vehicles Parking Lot with Vehicle-to-Grid Service in Microgrid” can help people know the importance of the grid stability. The proposed algorithm can also help paper “A hierarchical microgrid protection scheme using hybrid breakers” to locate the location of the breaker.
Reply: Thanks for reviewer’s advice. We modify the background part with the example given by you to prove the importance of identifying critical nodes. The modified background part is “The safe and stable operation of the power system faces challenges due to the continuous development, increasing size, and growing complexity of the modern power system and its grid structure [1-2]. In recent years, major blackouts, both do-mestically and internationally, have primarily resulted from the removal of critical nodes or lines due to failure or attack. This, combined with large-scale transfers of system power flow, leads to overloading or ultra-low-voltage operation of other nodes or branch circuits, causing chain failures in the power system [2-4].These nodes, vulnerable to greater disasters from failure or attack, are also referred to as critical nodes. Reference [ 5 ] analyzes the underlying causes of many global blackouts, many of which are caused by attacks on critical nodes in the power grid, which helps people understand the importance of maintaining power grid stability and identifying critical nodes. Therefore, accurately and effectively identifying these critical nodes in the power system is crucial for ensuring the safe and stable operation of the power grid. This effort holds significant theoretical and practical importance in preventing large-scale power outages in the grid.” All the changes in the manuscript have been highlighted in red color.
- In the comparison table, please also compare the latency with other algorithms.
Reply: Thanks for reviewer’s advice. Since the structure of the IEEE 39-bus system is not particularly complex, each algorithm can get the results in a very short time. Therefore, the impact of latency is very small in this case. However, we will further improve the algorithm in the subsequent research and apply it to more complex simulation systems and real power grids, and compare the latency with other algorithms.
- Please add more explanation for Fig.7. Especially the performance between the blue and green line.
Reply: Thanks for reviewer’s advice. In the process of revision, we added two illustrations in the previous text. Therefore, the original Figure 7 is changed to Figure 9, and a more detailed explanation is added to this figure in the manuscript.
The expanded interpretation is “Figure 9 illustrates that, there is a significant decrease in the load survival rate when the critical nodes are attacked by the IPRA-PFTE algorithm in the case of the load survival rate metric. the curve of IPRA-PFTE decreases faster and the final load survival rate reaches 25.61% when all the critical nodes are removed. After the first five critical nodes identified by the improved PageRank algorithm and Random Ma-trix & Entropy are removed, the load survival rate decreases slowly. This is because the two identify critical nodes from the perspective of node voltage data mining and the power flow distribution characteristics between nodes. The final load survival rates after all critical nodes are removed are 39.98 % and 52.81 %, respectively. ETPD-CNIA identifies critical nodes based on the output power of the generator, so its performance under this index is not as good as IPRA-PFTE. The final load survival rate after all critical nodes are removed is 34.35 %. IPRA-PFTE significantly outper-forms the other three algorithms under this metric both in terms of downward trend and final value.” All the changes in the manuscript have been highlighted in red color.
- Are there any drawbacks for introducing virtual nodes?
Reply: Thanks for reviewer’s advice. The introduction of virtual nodes is to make the PageRank algorithm more in line with the characteristics of the grid, the addition of virtual nodes will lead to too high an importance index of the virtual nodes in the results, so we remove the results of the virtual nodes in our study, and only retain the importance index of the original nodes in the grid.
References
[1] Tu, J; He, J; An, X; et al. Analysis and Lessons of Pakistan Blackout Event on January 23, 2023. Proceedings of the CSEE, 2023, 43(14): 5319-5329.
[2] Eisenberg, D. A; Park, J; Seager, T. P. Linking cascading failure models and organizational networks to manage large-scale blackouts in South Korea. Journal of Management in Engineering, 2020, 36(5).
[3] Lei, A; Zhou, J; Mei, Y; et al. Analysis and Lessons of the Blackout in Chinese Taiwan Power Grid on March 3, 2022. Southern Power System Technology, 2022, 16(09): 90-97.
[4] Yan, D; Wen, J; Du, Z; et al. Analysis of Texas blackout in 2021 and its enlightenment to power system planning management. Power System Protection and Control, 2021, 49(09): 121-128.
[5] Hu, Y; Xue, S; Zhang, H; et al. Cause Analysis and Enlightenment of Global Blackouts in the Past 30 Years. Electric Power, 2021, 54(10): 204-210.
[9] Liu, W; Zhang, D; Ding, Y; et al. Power Grid Vulnerability Identification Methods Based on Random Matrix Theory and En-tropy Theory. Proceedings of the CSEE 2017, 37(20): 5893-5901.
[15] Ma, Z; Liu, F; Shen, C; et al. Rapid Identification of Vulnerable Lines in Power Grid Using Modified PageRank Algo-rithm——Part I: Theoretical Foundation. Proceedings of the CSEE, 2016, 36(23): 6363-6370+6601.
[16] Ma, Z; Liu, F; Shen, C; et al. Rapid Identification of Vulnerable Lines in Power Grid Using Modified PageRank Algo-rithm——Part II: Factors Affecting Identification Results. Proceedings of the CSEE, 2017, 37(01): 36-45.
[17] Fan, B; Shu, N; Li, Z; Li, F. Critical Nodes Identification for Power Grid Based on Electrical Topology and Power Flow Dis-tribution. IEEE Systems Journal, vol. 17, no. 3, pp. 4874-4884, Sept. 2023,
[19] Li, C; Kang, Z; Yu, H; et al. Identification Method of Key Nodes in Power System Based on Improved PageRank Algorithm. Transactions of China Electrotechnical Society, 2019, 34(09): 1952-1959.
Thanks again for your valuable advice!

Round 2
Reviewer 3 Report
Comments and Suggestions for Authors
I did not see that the recommended papers for background are mentioned in the paper.
Comments on the Quality of English Languagenot bad